# Benefits of Tango Therapy in Alleviating the Motor and Non-Motor Symptoms of Parkinson’s Disease Patients—A Narrative Review

**DOI:** 10.3390/brainsci12040448

**Published:** 2022-03-27

**Authors:** Any Docu Axelerad, Alina Zorina Stroe, Lavinia Florenta Muja, Silviu Docu Axelerad, Dana Simona Chita, Corina Elena Frecus, Cristina Maria Mihai

**Affiliations:** 1Department of Neurology, Faculty of General Medicine, Ovidius University, 900470 Constanta, Romania; docuaxi@yahoo.com (A.D.A.); lavinia.muja@365.univ-ovidius.ro (L.F.M.); 2Department of Neurology, County Clinical Emergency Hospital of Constanta, 900591 Constanta, Romania; 3Faculty of General Medicine and Pharmacy, “Vasile Goldis” Western University of Arad, 317046 Arad, Romania; docu.silviu@yahoo.com; 4Department of Neurology, Faculty of General Medicine and Pharmacy, “Vasile Goldis” Western University of Arad, 310045 Arad, Romania; danaioncu@yahoo.com; 5Department of Pediatrics, Faculty of General Medicine, Ovidius University, 900470 Constanta, Romania; frecuscorina@yahoo.com (C.E.F.); cristina_mihai@365.univ-ovidius.ro (C.M.M.); 6Department of Pediatrics, County Clinical Emergency Hospital of Constanta, 900591 Constanta, Romania

**Keywords:** Parkinson’s disease, tango, rehabilitation, Argentine tango, therapy

## Abstract

The present study examines the efficacy of tango therapy on motor and non-motor symptomatology in Parkinson’s disease, as detailed in articles published over the previous four decades (1980–2022). All data was collected using PubMed, Google Scholar, Web of Science, and Science Direct. The present descriptive study outlines the advantages of tango in the rehabilitation of Parkinson’s disease’s motor and non-motor symptoms. Numerous studies have been conducted to determine the usefulness of tango for people with PD. Information from various research is critical for determining if tango is a useful supplementary therapy for the variety of symptoms related to Parkinson’s disease. The purpose of this review was to describe the present state of research on this subject. Thus, the objective of this review is to promote awareness of tango therapy’s therapeutic benefits for Parkinson’s disease.

## 1. Introduction

Parkinson’s disease is a degenerative neurological disorder that manifests with tremor, bradykinesia, and postural instability. Whereas the typical age at diagnosis is 60, the majority of those who are afflicted are in their 70s and 80s. The condition is caused by nigral and extranigral degeneration and manifests itself via a gradual decline in physical functions, cognitive and psychomotor abilities, and also emotional and mood disorders [1]. People with PD face a decline in their capacity to do everyday tasks, social exclusion, and a decline in their life quality [2].

Dance represents a multifaceted activity that stimulates the auditory, visual, and sensory systems, as well as provides opportunities for musical experience; social engagement; memory development; motor learning; and affective observation, expression, and interaction [3]. Numerous uncontrolled studies show that people with Parkinson’s disease are enthusiastic about attending dancing classes on a consistent basis, exhibit high compliance, and have a proclivity to continue participating after the study is over [4].

Tango contains various features of movement that might be particularly beneficial for those with Parkinson’s disease. When individuals dance tango, they are involved in a multitasking activity that needs dynamic balance and includes turning, initiating movement, and moving at a range of velocities that frequently reverse in close proximity to a partner.

Our analysis obtained articles from the following electronic scientific resources: PubMed, Google Scholar, Web of Science, and Science Direct. Between 1980 and 2022, pertinent English publications were discovered using the terms “tango”, “tango dance therapy”, and “Argentinian tango” in conjunction with “Parkinson’s illness” and “Parkinson’s disease symptoms”. One hundred articles were chosen through database searches. The search was manually supplemented with references from current studies and case reports. The titles of the documents were checked for accuracy, and duplicates were omitted. In total, 215 articles were identified; 31 articles were excluded after full-text screening; and 84 articles were excluded because they were duplicates. Publications from the scientific literature on tango and Parkinson’s illness met the eligibility criteria.

Reference lists from identified literature were manually searched for completeness by the authors to confirm content relevant to tango dance therapy in Parkinson’s disease patients. Studies were included in this review if they met the following eligibility criteria: (1) written in English; (2) type of article being an original article, review, or case report; (3) patients participating in the studies being diagnosed with Parkinson’s disease; (4) no external intervention during the experiment of the study involved; (5) a lot of patients with Parkinson’s disease only performing tango therapy as an alternative therapy at the time of the study; and (6) the patients with Parkinson’s disease included in the study to not be diagnosed with dementia and psychiatric disorders.

Studies were excluded if they met the following criteria: (1) citations and patents; (2) written in a language other than English; (3) abstract papers with no data; and (4) the alternative therapy to be any other type of dance therapy other than tango or Argentinian tango. The search results were analyzed, and the important material is given in this paper as a narrative review.

## 2. Parkinson’s Disease Motor Features

Usually, a person who is unhealthy is assessed medically. The body is seen as an “object” with quantifiable morphology, physiology, and pathology. Those metrics are used to determine if a person is “ill” or “healthy”. Typically, no extra evaluation of an individual’s perspective of the physique is made; nevertheless, this is necessary for a far more thorough picture of the body. The body is a personally perceivable reality with physically perceivable knowledge, feelings, and experiences in this situation. Past years have seen an explosion in raised attention and scholarly understanding of the relationship between the body and self-awareness. Body awareness, as defined by Mehling et al. [5], is recognized as the most critical common feature in body and mind treatment. Thus, awareness is permeated by the subjective, experiential part of body awareness and self-perception, which is malleable enough to vary as a function of cognitive functions such as attention, comprehension, judgment, ideas, recollections, conditioning, perspectives, and their effects. Additionally, owing to its effect on motor function [6] and its relationship to variations in motion patterns, body experience is critical for competence in everyday regular tasks [7]. Perceivable, intellectual, affective, or motor experiences of the body are all possible. Psychodynamic therapy and psychiatry are fields of study that focus on the impacts, modifications, and repercussions of (troubled) bodily experiences [8]. The unique relevance of bodily experience is in its role as a determinant of the experience of illness. It may manifest as a physical symptom (e.g., a disordered body image in individuals with disordered eating) or as physical sensations or judgments. Whereas somatic disorders are associated with pathologic changes in the organism and thus in the body experience, the study of body experience remains a small subfield within the field of physical diseases. For example, during an illness (body feelings), the sense of “being secure with one’s own body” or of physical well-being changes [9].

Patients with PD exhibit major alterations in their proprioception and other elements of bodily experience. It might have a detrimental effect on the individual’s well-being or might result in an estrangement from one’s own body, wherein one’s personal motor activity is seen as autonomous, correlating to a disruption in one’s body image. Individuals with Parkinson’s disease lose faith in their own bodily functions, which become progressively unreliable as the condition develops. Fright of falling (body sensation) and additional awareness of functional actions might induce additional tiredness and pain.

The most visible motor sign is the tremor. In the majority of instances, it is a resting tremor, unilateral, and most noticeable in the distal portion of the limb, and it subsides with movement or sleep. Rigidity is induced by an abnormally high rate of muscular activity [10]. As with tremor, it is frequently asymmetrical and is a common source of arthralgia, which could be a precursor to this condition [11]. Akinesia (or bradykinesia) is a term that refers to the sluggishness and difficulty associated with the entire movement process, including planning, initiating, and executing movement. This condition is particularly bothersome while doing everyday duties such as dressing or writing. Individuals with Parkinson’s disease typically have fewer issues with movement whenever an external signal is presented [12]. Postural instability and a tendency to fall are two more common and important symptoms of Parkinson’s disease that get worse over time [13,14].

Individuals with Parkinson’s disease often have more trouble turning during motion than they do while walking on a straight path, which puts them at a higher risk of falling. Such falls are eight times likelier to lead to a hip fracture compared to falls when walking normally [15]. Turning may also result in freezing during walking. Freezing, a frequent complication affecting 53% of individuals with Parkinson’s disease for more than 5 years [16], also happens at gait initiation and while going through doors or other restricted places [17]. Individuals with Parkinson’s disease frequently have trouble walking while doing two tasks [18,19,20].

## 3. Parkinson’s Disease Non-Motor Features

Dysfunction of cognition, thinking, emotions, behavior, and speech are all examples of nonmotor symptoms. Executive dysfunction, attention problems, slow cognitive speed, trouble remembering information you’ve already learned, and visuospatial problems are the most common types of cognitive impairments [21,22].

Whenever people with PD are required to walk while doing additional activities, including mental arithmetic, their gait speed, stride length, and gait stability all decline. Decreased functional mobility may result in reduced self-esteem, depression, withdrawal from activities, and a diminished quality of life [22].

The primary motor predictor of low quality of life (QoL) is established as balance impairment [23,24,25]. Falling, stumbling, and trouble rotating as a consequence of postural instability are found to be significant [26]. Around 70% of individuals experience falls, which can lead to devastating repercussions, including fractures [27] or stigmatization as a result of the shame associated with falling in public [26]. Fear of falling may prohibit individuals from participating in outside activities, thereby reducing their quality of life. Timed Up and Go performance (a straightforward test involving both dynamic and static balance) was shown to be a good determinant of QoL [28]. Balance enhancement is regarded as one of the most critical outcomes for determining if a treatment is effective and efficient [29]. Gait problems and unfavorable drug effects are two more motor issues that have an impact on overall life quality [30].

Tango, being one of the most investigated dance styles in terms of balance in Parkinson’s disease, is claimed to be quite beneficial for increasing quality of life. It places balance and gait training in an atmosphere conducive to social contact and intimate collaboration with a partner [31]. A non-randomized experiment of short and intense tango classes has shown that they greatly improve balance, but not gait [32]. In evaluating the benefits of tango and traditional exercise on healthy senior citizens and senior citizens with Parkinson’s disease, only the PD tango group improved all parameters of balance, falls, and gait in a randomized control trial research. Additionally, the PD tango group had greater balance confidence compared with the PD exercise group. Both groups expressed high levels of satisfaction with their tango practice and with the entire therapy [33,34].

## 4. Theoretical Improvements of Tango Therapy in Motor Features of PD

Music has been shown to stimulate certain brain circuits involved with emotion, suggesting that it may help to alleviate stress and improve social interactions [35]. In a randomized controlled trial of tango over 2 weeks (meeting 1.5 h and four times per week) with middle-aged adults who self-reported symptoms of anxiety and depression, massive changes in their symptoms were observed at the post-test and were preserved at the 1-month follow-up, and similar improvements were observed in the same cohort after 8 weeks of tango dancing [36].

Due to the fact that social dance frequently requires periodic gatherings to practice in couples or groups to learn skills and/or to enjoy the activity, dancing could foster collaboration, which could result in the development of social contacts and more community participation. Additionally, treatments that include self-expression, including emotional and physical, could be particularly beneficial for treating mental and motor symptoms associated with aging correlated morbidity. Treatments that enable patients to regain control of their bodies and engage in intensive social engagement have been found to improve quality of life [37].

In Parkinson’s disease rehabilitation, attention has shifted to related cortical regions in an attempt to correct for deficiencies in dopaminergic and non-dopaminergic circuits and to improve gait quality [38].

The argument would be that the striatum provides phasic signals to the supplementary motor area (SMA), which is capable of activating and deactivating every sub-movement inside the movement sequence, including for their efficient implementation [39]. In Parkinson’s disease, malfunction of the striatum may result in the loss of internal gait rhythm, resulting in their typical gait problems.

Sensory stimuli seem to be an effective method to enhance gait in Parkinson’s disease (PD) by compensating for dopaminergic, or maybe nondopaminergic, deficiencies [39]. Nevertheless, several forms of audio cueing may offer an external rhythm that compensates for the striatum’s dysfunctional inner rhythm in PD [39]. Additionally, some visual cues may aid in gait improvement. As a result, sensory cues may be used as a rehabilitation method for people with Parkinson’s disease [40].

Tango is a style of dance that incorporates external signals, movement tactics, and balancing exercises [41]. There is proof of enhanced putamen stimulation throughout regular, metric-rhythm movements, such as those seen in tango dancing [42]. Additionally, a change in cortical engagement in healthy tango dancers was observed, with greater efficacy in the premotor region and supplementary motor areas throughout the anticipated sequence of dancing movements [43]. Audio signals are considered to skip the striatum and directly enter the supplementary motor area via the thalamus or the premotor cortex via the cerebellum [43]. Tango could provide a substantial impact on displacing affected motor pathways in Parkinson’s disease. Movement problems are exacerbated in PD patients when they are doing a secondary activity concurrently, and it is well established that multitasking exercises may enhance their decreased motor performance. Certain dances, such as tango, provide an excellent opportunity for multitasking. Additionally, it has been shown that patients truly like this type of fitness routine, which has a high adherence (>90%), and realize that they are capable of doing activities further than the limitations imposed by Parkinson’s disease [43].

Consequently, in the past few years, the science world has shifted the focus of rehabilitation for individuals with Parkinson’s disease away from treatments with dubious results and toward approaches focused on cued tactics and cognition. The next step along this path was to recognize the value of mixing goal-directed physical exercise with emotional rehabilitation, paving the way for the adoption of non-traditional therapies including tango, especially Argentinian tango.

Tango dance can meet several of the essential categories listed as critical for a training regimen tailored for people with Parkinson’s disease [44]. To begin, dancing is a musical practice that may serve as an extrinsic trigger to increase the initiation and quality of motion, which is a critical component of a Parkinson’s disease rehabilitation program. Furthermore, people must concentrate on their partner’s motions, complete synchronization, stepping tactics, and motion aesthetics [45].

Dance, in particular, could be useful for those with Parkinson’s disease. Current data indicates that basal ganglia, the areas most afflicted by Parkinson’s disease, are engaged, especially in the regulation of dance movements. Brown et al. [42] utilized positron emission tomography to investigate the brain areas associated with the regulation of tango movements of a particular lower limb in supine healthy people. While tango motions were done to a metered beat in a predictable pattern, they observed significant growth in the basal ganglia, notably in the putamen. At the moment, the usefulness of this information for people with Parkinson’s disease is unknown. Because tango motions stimulate basal ganglia function, the advantages of tango, like most dances, need movement coordination to a rhythm. Utilization of auditory cues to aid mobility is recognized to be useful for patients with PD.

Apart from the precise moves, tango, like other dances, requires synchronization of movement to music. External acoustic cues could be provided through music. Use of these signals to aid movement has been shown to be advantageous for those with Parkinson’s disease [46,47,48]. External cues have the potential to contact cortical circuitry, bypassing the malfunctioning basal ganglia [49].

Additionally, training using auditory cues may help alleviate the intensity of freezing [50]. Auditory signals may be able to circumvent the faulty circuit connecting the striatum with the supplementary motor region through the thalamus, which is generally employed for internal cued movements [51]. Auditory signals might contact the premotor cortex through the cerebellum, according to research [52]. The utilization of rhythmic signals from music could therefore be a critical aspect of dancing as a therapy for people with Parkinson’s disease. Indeed, music therapy has been shown to enhance motor function, daily living activities, mood, and life quality in people with Parkinson’s disease [53].

Various research have shown the efficacy of various techniques and revealed beneficial benefits on motor function, health-related quality of life, balance, leg strength, postural instability, bradykinesia, and walking [54]. While regular participation in physical activities is required to produce beneficial treatment outcomes, people with Parkinson’s disease frequently lower their exercise intensity due to reduced mobility, concern of falling, or poor achievement goals. Along with mind–body medicinal techniques to exercise, including Qi Gong or Tai Chi, dancing has been suggested as a suitable intervention [55]. Music-based movement therapy for Parkinson’s disease patients naturally integrates cognitive movement tactics, cueing techniques, balancing exercises, and physical exercise [56]. This may provide a greater incentive for long-term engagement than standard fitness training.

Along with medication, physical therapy has been shown to be useful in controlling the symptomatology of PD. Traditional physiotherapy, treadmill training, cueing, techniques for elaborate motor sequences, massaging, martial arts, and dancing have all been found to be useful in regaining stability, and life quality, at least temporarily [57,58].

Tango dancing could face various management problems, such as learning tango teachers to develop in a healthcare setting, locating an adequate location for dancing, planning an opportune setting for every attendee to participate, and tolerating potential time delays due to the respondents’ mobility impairments. A feasible way to circumvent the limits outlined before would be to create a system that can be completed from home and yield comparable results. By completing motor exercises at home according to defined protocols, it is possible to maintain the continuation of the rehabilitation program, leading to lower national health expenses. Various physical-therapy treatments using “at-home” regimens for individuals with Parkinson’s disease have been suggested, including a public fitness routine [59], kinetic-based physiotherapy [60], and a customized occupational therapy plan [61,62]. Every one of these strategies has been demonstrated to be healthy and to increase self-confidence in everyday tasks.

Tango treatment offers a high level of patient compliance since it is conducted to music and participants move in time with the music. Tango treatment, for instance, is an excellent treatment method for correcting the features of Parkinsonian gait with freezing and festination since the patient is instructed or encouraged to conform to the music’s regular rhythm. Furthermore, tango treatment is assisted by music and has the therapeutic advantages of music. The tango method, in many respects, is much more than a dance. The Tango method is also a kind of physical therapy, since tango motions are composed of many basic tango moves that engage the majority of the muscle system. Thus, tango has the impact of multiple physical treatment techniques being applied concurrently to the entire body, which is not possible with any other kind of physical therapy now available.

## 5. Theoretical Improvements of Tango Therapy in Non-Motor Features of PD

Parkinson’s disease is the second most prevalent neurological illness among adults over the age of 60. A dysfunction of the basal ganglia results in serious motor and non-motor symptoms. Additionally, the condition is linked with major psychosocial impairments and a decline in quality of life [52]. Non-motor symptoms and psychosocial factors have a substantial impact on quality of life [52]. Given the demographic changes now occurring in the western industrialized world, it is reasonable to anticipate a significant rise in PD during the next several years. Although there is widespread recognition of the relevance of non-motor symptoms and psychosocial variables, therapy for Parkinson’s disease has traditionally concentrated only on motor symptoms. It is one of the causes behind the present rise in popularity of “holistic” alternative treatments. According to a 2001 study of US clinics, around 40% of people with Parkinson’s disease might benefit from alternative therapy [63].

Regarding conventional PD treatments, it is known that while dopamine-receptor agonists or deep brain stimulation might alleviate PD symptoms, they might well have adverse consequences, including drug-induced dyskinesia and medical complications [64].

There has been a surge of interest in non-pharmacological therapy in recent periods, particularly the impact of movement in general, and dancing in particular, on people with Parkinson’s disease. Various recent reviews [45,65,66,67] document dance’s impacts on physical functioning and also cognitive and psychological results, including depression level, pleasure, and well-being.

Dance has been extensively studied in senior individuals, demonstrating that this relatively low-impact type of physical exercise improves balance and cognition [68].

Apart from its impacts on gait and balance [69,70,71,72,73,74], tango has been shown to improve quality of life [4], as well as individual and interpersonal activities [75]. Other observations are supported by a survey conducted by Quiroga Murcia et al. [76,77] on the advantages of dancing among adults; respondents described how dancing improves emotional and physical areas of health, and also spiritual and social domains, most notably self-esteem and coping techniques [77].

Meta-analyses and review articles [41,78,79,80] all concur that Tango had the greatest influence on enhancing gait, balance, and QoL for people with Parkinson’s disease.

Recreational dancing in a social context has been shown to increase motivation in older people [81]. Patients over the age of 65 who have shown sustained engagement in social dancing had improved balance and gait function compared to age-matched nondancers [82]. Activities that interest and attract older adults are necessary, since roughly 60% of Americans over the age of 65 do not meet the daily recommended level of physical exercise [83]. Patients with PD had even lower levels of activity, around 15% less than age-matched controls [84].

Tango lessons’ social contact, social support, and social influences very certainly had a good impact on participation as well. In a group context, behavioral models, the formation and enforcement of social norms about health-promoting conduct, and the development of social networks would all be possible in a group context [85]. Indeed, multiple studies found that members of the tango group engaged in social activities outside of class, such as attending concerts, the symphony, and social dances. On an individual basis, the involvement of a companion may also have aided people with PD in getting more confident pushing oneself in the challenges inherent of movements [45], therefore facilitating mastery experiences, a main source of self-efficacy [86]. The increases in self-efficacy that might happen during the tango classes may have carried over into everyday life, promoting the interest in participating in additional or novel practices, reattempting previously abandoned activities, or dedicating the motivation and engagement needed to sustain one’s existing activity level.

Patients with Parkinson’s disease who were motivated to experiment with alternate movement methods using dancing showed improvements in neurological state and movement initiation [87]. Tango, like other treatments, necessitates balance and an awareness of motion control. Tango, on the other hand, is distinct from other complementary movement methods because it is conducted with a companion in a context that encourages social inclusion; it is also dynamic by the fact that the attendee is constantly learning; and finally, it is executed to music that can interact with the attendee in addition to acting as an exterior signal.

In healthy individuals, dancing has been shown to enhance alertness shortly afterwards [88]. On either hand, the research is devoid of research examining the repercussions of tango dancing on people with Parkinson’s disease, apart from a single case study examining the effects of transcranial direct-current stimulation on core motions and gait throughout tango dancing [89].

According to subjective experiences of patients, dancing seems to be a beneficial and joyful therapeutic exercise for older and physically challenged individuals in terms of physical, mental, and emotional well-being [90]. It was also found that dancing improves quality of life, increases program compliance, and improves cardiovascular health in adults with chronic heart failure [91].

Participating proactively in the formation of compensating actions resembles the sensation of recovering bodily control deliberately. This was seen by participants as an improvement in communication connecting “body and mind”, which is especially important for those with PD who suffer a lack of autonomy of their motor abilities as a result of motor symptoms, impairing their quality of life [92,93]. As a result, it can be hypothesized that active engagement in the formation of compensating motions can promote an effectively experienced and problem-solving coping behavior that correlates favorably with an increased feeling of well-being. One aspect that could aid in the development of new motions and body control is the acknowledgment of one’s own limits.

Concerning biomechanical results, there is a shortage of published results owing to the fact that, although there has been usage of kinematic parameters, biomechanical measures were just lately included in a collection of reference index data for Parkinson’s disease patients [94]. Considering these facts, the most intriguing biomechanical modification seen after 4 weeks was an enhancement in pelvic rotation in 3 of 10 individuals. The development might be attributed to the elevated posture enforced by tango dancing.

## 6. Studies from Literature

Findings suggest that dancing may potentially be utilized therapeutically to successfully target balance and complicated gait problems in healthy older persons [95]. Among senior persons, dance/movement therapy has been advised to help them improve or maintain their range of motion [96]. Dance/movement therapy has also been shown to be an effective therapeutic approach for people with Parkinson’s disease. Advancements in movement starting were seen in a group of patients with Parkinson’s disease who engaged in free-form movement [87].

According to the American College of Sports Medicine, the advantages of intense exercise may be obtained with three to five 20–60 min sessions each week. Short, intense bursts of tango dancing that achieve these qualities may represent an effective kind of endurance training, according to the same recommendations: 1.5 h a day, 5 days per week, for 2 weeks may enhance balance, gait, and mobility. Despite the rigors, patients stayed enthusiastic, and one experiment had a drop-out rate of just 14%. This rigorous program had the added benefit of surpassing 180 min per week, the minimal amount of exercise required for various forms of exercise to have a substantial influence on habitual gait parameters in older adults [45].

Researchers have observed improvements in patient posture, walking, speech, and UPDRS items, indicating that this type of dancing could have an effect on axial symptoms, particularly in PD. The advantages reported following regular tango classes are comparable to those reported following this home programme [97], and they may be attributed to the impacts of the music’s particular tempo [98], to the creation of motor techniques throughout dancing [99], and to social/psychological impacts [66].

Nevertheless, scientific data demonstrating the efficacy of complementary therapy in people with Parkinson’s disease is quite scarce. Kwok et al. conducted a meta-analysis in which they incorporated nine pieces of research on the impact of various alternative treatments into a quantitative review [100]. Tai Chi, dancing, and yoga all had a substantial favorable impact on functional mobility and motor clinical signs as judged by the Unified Parkinson’s Disease Rating Scale (UPDRS).

A systematic review and meta-analysis of studies on tango and Parkinson’s disease [65] demonstrated that it had highly significant favorable impacts on motor severity, as evaluated by the Unified Parkinson’s Disease Rating Scale (UPDRS), and on balance and gait. One mechanism through which tango could enhance postural techniques in patients with Parkinson’s disease is through the stimulation of complementary motor areas via a motor imagery effect, such that simply training to visualize tango dance results in a growth of active bilateral motor areas throughout locomotor imagery [101]. Additionally, tango dancing enables PD patients to design and then perform novel dance schemes that synchronize their trunk and limb motions, resulting in the recovery of executive cortico-subcortical output [32].

The commonly used life quality assessment for persons with Parkinson’s disease, the PDQ-39, assesses the bodily experience solely indirectly via mobility, daily competency, and physical complaints. Schrag et al. [62] showed that people with PD performed markedly lower on such scores than those without PD. Holmes and Hackney [102] conducted an investigation to ascertain the perceived effect of participating in a modified tango class on the quality of life of 16 people with Parkinson’s disease. The findings imply that when delivered in an organized setting with expert training, modified tango can enhance abilities for everyday tasks and lead to an improvement in QoL. The research examined body awareness, body control, social advantages, and dissatisfaction inside the context of quality of life, but did not concentrate on patients’ bodily experiences.

An intense study of contemporary dance for 11 persons with initial phase Parkinson’s disease similarly showed improvement on the Fullerton Advanced Balance Scale but not on the Timed Up and Go test [103]. When the effects of partnered and unpartnered dancing on balance and mobility were compared, both groups demonstrated increased balance and walking ability. Nevertheless, exclusively coupled participants showed greater satisfaction with lessons and a desire to continue [45]. Although dance events have been shown to enhance movement patterns in PD patients, uncontrolled research found no change in the Timed Up and Go and semitandem tests. While dance therapy has been shown to enhance QoL not only in patients, but also in caregivers [104], it also has a favorable impact on participants’ attitudes, reduces apathy and despair, and enhances neuropsychological function [105,106]. Besides the physical advantages, Parkinson’s disease (PD) patients who attend dance lessons claim socio-emotional advantages as well [107].

Similarly, the majority of research on Parkinson’s disease patients employed semi-quantitative ratings from specialized assessment scales (for example, the Berg Balance Scale and the Timed Up and Go test), while just a handful used three-dimensional gait analysis. Hackney et al. reported an increase in the proportion of time spent in stance ortostatism during forward walking [39] and cadence [45]. Therefore, among all the spatiotemporal factors, the cadence is perhaps the best indicator of tango’s impact, which is presumably owing to the melodic rhythm’s impact.

According to Aguiar et al. [108], dance therapy might improve the mobility and quality of life of patients with Parkinson’s disease. Numerous studies indicate that Argentine tango could have a rapid training impact on motor symptoms in those with Parkinson’s disease. Lötzke et al. [69] review summarized the findings of the research and discovered a considerable increase in motor function. Nevertheless, no considerable reduction in the incidence of the freezing gait symptom was detected; also, no substantial change was seen using the Six Minute Walk Test in four considered investigations. The findings are similar to research on the effect of dance therapies on motor symptoms associated with PD generally. For instance, Sharp and Hewitt’s [80] meta-analysis revealed that dancing generally improves UPDRS motor symptoms, equilibrium as evaluated by the Berg Balance Scale, and walking speed.

In a study by Rawson et al. [109] that examined tango, treadmill walking, and stretching, they reported that only the treadmill cohort increased forward and backward velocity, but not the tango group. Further research examined tango to tai chi as an active control and found no significant improvement in the quality of life or subjective wellbeing of PD patients [110].

Nevertheless, the majority of research focuses on motor symptoms, with few examining the impact on well-being and non-motor symptoms [100]. There is evidence that TA has a beneficial impact on tiredness and active engagement in moderate-intensity exercises [34,75]. Numerous research studies [34,65,111] and individual situation research have demonstrated a beneficial effect on wellness quality of life [HRQoL], as evaluated by the PDQ-39. Hackney and Earhart [112] showed a substantial influence on the PDQ-39’s mobility, peer assistance, and PDQ score. The above impact was seen following tango dancing in 2009, while no impact on HRQoL was observed following Tai Chi or Foxtrot.

Hackney and Earhart [112] discovered that a 10-week dance program increased active balance, gait velocity, and cadence in adults with mild to severe Parkinson’s disease. This was significant for both non-partnered and paired dancing, although the paired dance group reported more satisfaction and desire to continue the program.

McKee and Hackney [111] demonstrated that 12 weeks of community-based Tango classes enhanced spatial cognition, balance, and executive functioning, while also reducing illness severity when compared to the controls undergoing educational sessions.

McKee et al. [112] and Romenets et al. [34] were unable to replicate these findings. Two research studies that examined the impact of tango on depressed symptoms and apathy constantly found no significant differences [34,75]. Additionally, TA’s impacts on cognitive function were equivocal. While McKee et al. [111] discovered a considerable shift in the spatial imagination’s mental function, Romenets et al. [34] discovered just a tendency in this shift. The tendency toward improved mental function is bolstered by the findings of recent research conducted by DeNatale et al. [113], which discovered a considerable increase in mental function (attention and executive functions).

According to Holmes and Hackney [102], research involving 16 people with Parkinson’s disease demonstrated tango might “enhance abilities for involvement in everyday activities” and, in this way, enhance the standard of living. Zafar et al. [114] observed that modified tango enhanced “some areas of engagement”, most notably social life. Albani et al. [115] evaluated 4-day home workouts combined with tango dancing lessons over a 5-week period and observed improvements in motor performance, kinematic function, and life quality. Koch et al. [116] observed that one (90-min) tango session improved patients’ well-being and bodily self-efficacy while also improving the quality of their moves and other aesthetic appeal. Therefore, although some data indicate that various life quality markers might increase, little is documented about how TA therapy can affect the bodily experience of PD patients.

McKee et al. [111] reported that 23 persons with mild-to-moderate PD engaged in 30 h of customized tango and showed improvements in balance, spatial cognition, executive function, and disease severity when compared to a control group. At 10–12 weeks post-treatment, the tango group demonstrated improvements in illness severity and mobility, with benefits maintained.

In McNeely et al. [117] paper, Parkinson’s disease patients participated in tango dancing and a class structured after the Dance for PD program. Both patient groups showed comparable improvements in balance and mobility measurements. Additionally, the tango intervention group had more significant improvements in motor sign severity and functional mobility.

According to Peter et al. [118] paper, the tango group with Parkinson’s disease exhibited a substantial decrease in fall risk when compared to the control group. Additionally, tango dramatically enhanced gait performance in their limited study.

According to the Romenets et al. paper [34], Argentine tango may benefit patients with Parkinson’s disease in terms of balance, functional mobility, and satisfaction with treatment, with perhaps small advantages for cognition and fatigue.

Seidler et al. [119] examined group tango instruction via telerehabilitation in their paper. Twenty-six individuals with mild to severe Parkinson’s disease were randomly assigned to a Telerehab or an in-person training group. Both groups met twice each week. Both groups improved considerably in terms of balance and motor symptom severity over time.

Telerehabilitation, or the delivery of rehabilitation therapies remotely using telecommunications, has been explored lately as a method of overcoming hurdles and increasing compliance in a variety of patient groups, with promising results, especially being beneficial in the pandemic period [120,121,122]. Telerehabilitation techniques resulted in gains equivalent to in-person therapy and high engagement with a walking program for patients with Parkinson’s disease [123,124].

Nevertheless, other studies consider the potential disadvantages of tango, such as the demand to learn precise moves and the fact that certain patients may be uncomfortable dancing with the technique required by tango. Researchers recommended a more improvised style of dancing that might give a more approachable technique while retaining the utilitarian and social aspects of tango. Contact improvisation dance shifts the dancer’s emphasis away from apparent presentation and exact movement patterns and toward nonverbal communication and feeling. Contact improvisation dance tests an individual’s ability to respond to unexpected movement caused by tactile connection with a dance partner. In one uncontrolled trial, contact improvisation dance demonstrated statistically significant benefits on a variety of balance measures and tests. Furthermore, PD dancers stated a high level of satisfaction and a decision to join future contact improvisation dance lessons [125].

### Argentine Tango

Argentine tango (AT) exposes people with Parkinson’s disease to the unique restrictions imposed by movement and physical impairments, resulting in a more conscious perception (and ‘neutral’ acceptance) of their motion shortfalls. That can be linked to poor feelings of irritability and traumatic events, particularly at the start. Nevertheless, AT creates a basis within which dancers may create corrective or compensating motions that result in enhanced mobility and pleasurable bodily feelings while dancing.

There are a variety of methods wherein Argentine tango may have a beneficial effect on activity engagement in people with Parkinson’s disease. As a kind of physical activity, it assists individuals with Parkinson’s disease who have movement difficulties and might delay disease development [73]. That might lead to higher capability for everyday function and involvement. Notably, advanced tango sessions include characteristics that might improve participation in a way that regular exercise does not. Tango, for instance, needs working memory, attention management, and multitasking in order to combine recently taught and recall dance parts, maintain rhythm with the music, and move around other dancers on the dance floor. Self-initiated motions and motor planning are required for lead, whereas analyzing and reacting properly to the leader’s physical cues is required for follow [126]. Such cognitive difficulties can enhance everyday effectiveness and lead to higher or sustained engagement.

AT may also serve as a substitute for more standard physical education modalities. Supposedly, this kind of dancing may be especially beneficial for people with Parkinson’s disease, since it presents combined physical and cognitive demands. Additionally, the aerobic character of the activity, even at a modest intensity, might result in a certain increase in exercise intensity [73]. A randomized controlled study of community-based dance examined a cohort of Parkinson’s disease patients who attended 1-h tango sessions twice per week for 1 year compared to a group of PD patients who did not get any exercise intervention. Motor symptoms decreased significantly in the tango group but remained constant in the control group. The tango group’s stability and gait ratings were much higher and increased with time. Additionally, balance deteriorated in controls with time, but it enhanced in the tango group; no variations in freezing of gait were seen between groups, but the 6-min walking distance improved in the tango group and dropped in the controls. Upper extremity function increased markedly in the tango group in comparison to the control group. Furthermore, there were no considerable differences in everyday activities across the groups [73].

Numerous studies have shown that Argentine tango, a music-based dance therapy, is effective for improving functional ability, life quality, equilibrium, lower extremity power, postural control, bradykinesia, and locomotion in people with PD [34,65,127].

Individuals should concentrate on stepping technique, motor control, and sensorial awareness in Argentine tango. Also, dancers should observe their partner, course of motion, surrounding dancers, and the quality of the dance [112]. When learning Argentine tango, learners understand how to create with spontaneous emotions, steps, and movements. In comparison to other dances (such as the waltz or foxtrot), the Argentine tango features a broader variety of rhythms [4]. Given the absence of negative effects, Argentine tango can be recommended to patients with PD [111].

Argentine tango is correctly reported in research examining the intervention’s efficacy and effectiveness in persons with PD. The research, on the other hand, may be improved. The therapy influences the key regions that physiotherapy targets in patients with PD [128]. The Argentine Tango dancing program, which was created to fit the talents and balance constraints of persons with PD, is the most studied and effective intervention currently available.

Recently, there has been an increase in debate among academics and dance artists on the benefits of Argentine Tango as a music-based movement therapy for people with Parkinson’s disease [79,129]. At the same time, AT can increase spatial cognition since participants can acquire spatial postures and basic pathways throughout dancing courses, which must be retained, recalled, and re-used [111]. Nevertheless, patients are not required to remember or follow intricate step sequences; instead, it is critical that people learn to freestyle with natural responses, steps, and motions to the music. In contrast to other dances that include less rhythmic variation (e.g., the Waltz or Foxtrot), AT incorporates rhythmic variety [4].

Participants that engage in AT should concentrate on a variety of factors, including the partner’s motions, entire synchronization, stepping techniques, and the aesthetic elements of movement [112]. Additionally, tango utilizes external inputs that might result in more fluid movements [130]. Additionally, physicians could be inclined to recommend AT to PD patients due to the absence of adverse effects [111].

AT may significantly enhance the quality of life of persons with Parkinson’s disease by easing more than just physical symptoms [131]. Providing strategies that enhance patients’ social networks and also boost their self-esteem could be critical for promoting emotions of well-being in people with Parkinson’s disease [132]. Once patients feel a sense of accomplishment as a result of learning specific dancing skills and their dancing partner stays in the course, participants can notice an increase in self-efficacy, self-esteem, and enjoyment. Additionally, McNamara demonstrated the importance of individual and family relations in life objectives for patients with PD [133]. Nevertheless, the majority of present rehabilitation programs place little or no emphasis on this subject [133]. Dancing with a partner has been shown to improve social and personal interactions whilst still improving physical limitations such as axial impairments and dynamic stability [31]. Every move in AT is slow and close to the dancing partner. He/she gives stability by assisting those with PD who are uneasy due to their unstable and motor affections with their balance via the use of their body. Furthermore, AT can speed the acquisition of motor skills [45,126].

Jacobson et al. [134] revealed early findings on the effects of Argentine tango classes on clinical parameters of balance and gait in fragile older adults compared to walking. Researchers observed that the tango group improved more in terms of balance and difficult gait tests than the walking group. This shows that tango’s movements and sequences could be just as beneficial as strength/fitness training in correcting balance and gait problems.

AT might just have a beneficial effect on the body experience of people with Parkinson’s disease through increasing body awareness, body control, and acceptance of illness loads. Body awareness could be critical as an integrated activity for improving body experience in people with Parkinson’s disease. TA provides a framework for dancers with Parkinson’s disease to produce corrective or compensating motions that result in enhanced mobility and good bodily feelings while dancing. According to the practicality of new dance techniques and tango walking, they may indeed be intentionally practiced in daily life.

Duncan et al. [74] found that the increases in the MDS-UPDRS I score, which represents decreased nonmotor symptom severity, for the AT group imply that dancing can benefit not just motor symptoms of Parkinson’s disease but also nonmotor symptoms. Over a 2-year period, participation in community-based AT sessions improved motor and nonmotor symptom severity and balance in a small cohort of adults with Parkinson’s disease. Their findings reveal a reduction in the severity of Parkinson’s disease and improved physical functioning in a randomized experiment comparing a tango intervention group to a control group.

Also, Duncan and Earhart observed a comparable rise in Mini-BESTest scores in persons with Parkinson’s disease following a year of AT [73]. At 24 months, the AT group displayed improvement in forward and backward walking velocity relative to baseline, but the control group had a drop in velocity. Substantial improvements in forward walking velocity and nonsignificant increases in backward walking velocity have been previously described following participants’ dance for 1 year [73]. Additionally, the AT group displayed decreases in forward and backward walking velocity from 12 to 24 months, indicating that such benefits may have been lost over time or that the intensity of the AT dancing class was inadequate to sustain the advantages reported at 12 months.

Hackney and Earhart [4] discovered that AT, due to its improvised character, is more beneficial in enhancing gait, balance, and quality of life in people with Parkinson’s disease than other dance lessons. Parkinson’s patients might not be aware that they are walking backwards and that they can also turn firmly backwards and forwards. Patients may still not notice they are going backward because they are so engaged in following and tuned into their partner in the moment.

## 7. Results

As a result of our review, it was observed that tango is beneficial in Parkinson’s disease for reducing motor symptom severity and freezing of gait, and improving balance, gait, endurance, and upper limb mobility. Furthermore, the non-motor symptoms have been beneficially influenced by tango regarding the reduction of fatigue, enhancing the quality of life, participation, and clinical global impression of change. Tango, in addition to medical therapy, is a useful tool in improving the lives of Parkinson’s disease patients, regarding both motor and non-motor symptoms, not to mention the other non-Parkinson’s-disease-related advantages of using this therapy. Table 1 is presenting the studies for the beneficial evolution of both motor and non-motor features of Parkinson’s disease after tango therapy.

Our narrative review is valuable because it summarizes the literature and provides direction, given that the sources are of appropriate methodological quality. Our narrative review expands the body of knowledge on the subject of Parkinson’s disease therapy through tango and its benefits being summarized after the literature citations were chosen to be appropriate and balanced.

Tango therapy is very successful because of the systematic examination and application of fundamental tango components. Furthermore, tango treatment is neither a straightforward physical therapy nor a straightforward dance. Until today, it was commonly assumed that the advantages of tango therapy were psychological in nature. On the contrary, medical records indicate that the purpose of tango treatment is physical rehabilitation. The neuromuscular dysfunction is the most serious medical condition associated with Parkinson’s disease that is alleviated in practicing tango. Tango therapy, on the other hand, has been found to have psychological positive benefits and has been used in psychiatric domains. Tango therapy provides both physical and psychological benefits for patients suffering from any condition, and it is more than a dance.

Tango therapy is indeed a confluence of different physical therapeutic interventions, as well as music and cognitive therapy. Tango therapy’s results, nevertheless, appear to be greater than the total of numerous physical treatment modalities.

As our review examined articles published in the literature over a four-decade period, we noticed that studies conducted from the 1980s to the 2000s were modest in coverage, given the materials and methods used, which included more inventories and case reports, and we also noticed that the studies mentioned used subjectivist (related more to the answers of the patients and less on the neurological assessment) pre- and post-test scales and questionnaires as measuring instruments i.e., [38,87,90]. Additionally, in relation to the aforementioned studies, we observed a dearth of scientific foundations for comparison, particularly in the introduction and discussion chapters. Because this is a relatively new field of study, studies conducted prior to the 2000s lacked the necessary foundation for analyzing the findings of articles in comparison to other publications in the literature to ascertain similarities or differences in the results obtained. Another feature that is more or less accurate is that research conducted in the beginning of the studied period had a more critical stance toward the issue due to a lack of evidence in the literature from that period.

Controversies in this field of scientific study and on this subject arise from the belief that tango had more of a placebo effect than an apparent impact on motor symptoms, a fact reinforced by the use of self-assessment scales throughout the first investigations. Additionally, the therapeutic implements for Parkinson’s disease patients with non-motor symptoms were discovered later and initially received less attention because the non-motor symptomatology of Parkinson’s disease patients was not adequately considered, and the effects of non-motor symptomatology were not discovered to have such a significant impact on the patients’ quality of life. Another point of controversy stems from the fact that recovery through exercise, physiotherapy, and kinesiotherapy is extremely difficult and the effects decrease over time without consistent and continuous practice in this chronic and progressive neurodegenerative disease, whereas tango practice has demonstrated rapid and beneficial effects (after more than 20 tango lessons) sustained for an extended period of time. The comprehensive investigations on the subject of tango benefits in Parkinson’s disease demonstrated the effectiveness of this alternative therapy over a considerable length of time of extensive studies.

## 8. Further Directions

Existing research shows that tango is a beneficial method for people with Parkinson’s disease and positively affects PD-related symptoms. Additionally, significant improvements in various motor and non-motor complaints have been seen. The impacts of tango on interpersonal interactions and psycho-emotional well-being have received relatively little attention. Additionally, the use of tango as a motivator in the treatment of patients with PD should be explored further. Scientific investigations should include a larger sample size and concentrate on long-term impacts.

The current research provides preliminary evidence for the usefulness of community-based dancing in alleviating the motor and non-motor symptomatology among people with Parkinson’s disease. Future research might examine the nature of the changes in involvement that are happening, as well as the overall relevance of the many dimensions of engagement to general health and well-being in people with Parkinson’s disease.

We believe that more clinical trials on this subject are obviously needed. Randomized control studies must be developed and done together with neurologists, sports doctors, and investigators. Specific trial designs should be used to evaluate the potential disease-modifying benefits of tango therapy. Larger controlled studies assessing dance in Parkinson’s Disease and moreover other types of dance therapies are required to establish if certain dance intervention characteristics including dance style, abilities trained, amount of exercise, and class length, are best suited for targeting specific symptoms in people with Parkinson’s disease. Additional research with active control groups would be beneficial to determine tango’s unique contribution in comparison to other exercise programs.

## 9. Limitations

Our narrative review primarily focused on the conclusions reached in various studies on the subject. The limited number of participants in each of the included studies is a significant limitation. Another limitation is he difficulty of identifying and integrating complicated connections that may occur across a vast number of research. Additionally, this review included studies involving patients with mild-to-moderate severity of illness. Studies including individuals with varying degrees of disease severity could enable investigators to provide more personalized content and focus attention on whether tango gains in motor sign severity, life quality, equilibrium, mobility, and gait change throughout the illness’s progression. The potential of making judgments that are misrepresentative is because of the selection bias and subjective assigning of the research selected for the evaluation. Also, comparing research literature is complicated by variations in research design, such as the program variables utilized including class duration, regularity, and period of the therapy; the kind of activity practiced; and the results assessed.

There are not enough dance investigations in the PD research that make comparisons of dance therapies explicitly. To make recommendations to persons with PD about which activities would effectively address their symptoms and limitations while also promoting compliance, future research with bigger sample sizes will be necessary to evaluate different exercise programs.

## 10. Conclusions

In summary, the literature reviewed confirms the effectiveness of a home-exercise tango program and adds additional evidence to validate the possible usage of tango to enhance motor function in patients with Parkinson’s disease. The findings of this study imply that dancing could be an efficient, entertaining, and effective approach for controlling the symptoms of Parkinson’s disease. While home-based exercises would not be a reasonable replacement for certain facets of a live lesson, including group interaction and tactile and motor responses of another person, a residence physical therapy method utilizing tango might just have favorable impacts on posture and gait, and therefore a decrease in the financial impact of Parkinson’s disease.

While delivered in an organized atmosphere with expert training, adapted tango can gradually enhance mobility and confidence for participation in everyday activities, hence leading to an improved quality of life for those with Parkinson’s disease. These results highlight the importance of the complete care of Parkinson’s disease, which should incorporate occupational, somatic, and cognitive therapies, as well as the possibility of including dance disciplines. Much more is the possibility of establishing persuasive relevance via kinematic data and, eventually, clinically relevant information, including the effect on motor oscillations and so-called non-motor symptoms.

Academic publications form the foundation for implementing evidence-based clinical practice. Explicit details of intervention strategies in articles are a necessary but not sufficient condition for promoting innovation in medical practice and the development of successful treatment strategies. Earlier research has shown a deficiency in the documentation of non-pharmacological therapies. The characterization of physiotherapy treatments is especially difficult owing to the treatment’s many and complex parts.

Dance treatments in clinical practice may help patients maintain elevated levels of daily mobility and social activity by providing joyful and attractive experiences through and with their bodies.

We believe that tango dancing, as a multitasking activity that incorporates auditory, visual, and sensory signals, provides a pleasurable method of rehabilitation for Parkinson’s disease patients by enabling them to engage with motion via the melody of music.

## Figures and Tables

**Table 1 brainsci-12-00448-t001:** Representative studies for the beneficial evolution of both motor and non-motor features of Parkinson’s disease after tango therapy.

Effects	In Comparison with	Scale Used	Duration of the Study	Author	Bibliography
Motor symptom severity					
Improvement in motor symptom severity after practicing tango	Control group—no intervention	The motor component of the Movement Disorder Society United Parkinson’s Disease Rating Scale Part III (MDS-UPDRS-III)	1 year	Duncan et al.	[73]
Improvements in motor symptom severity, improve balance, and functional mobility after practicing tango	Physiotherapy group or exercise group	The motor component of the Movement Disorder Society United Parkinson’s Disease Rating Scale Part III (MDS-UPDRS-III)	12 weeks	Rios Romenets et al.	[34]
Stagnation for the symptomatology of the tango group, and worsening in the no intervention control group	Control group—no intervention	The motor component of the Movement Disorder Society United Parkinson’s Disease Rating Scale Part III (MDS-UPDRS-III)		Hackney et al.	[4]
Balance					
Improvements in balance among tango group	Control group—no intervention	Mini-Balance Evaluation Systems Test (Mini-bestest) of Dynamic Balance	1 year	Duncan et al.	[73]
Improvements in balance among tango group	Control group—no intervention	Mini-Balance Evaluation Systems Test (Mini-bestest) of Dynamic Balance	2 years	Duncan et al.	[74]
Improvements in balance among tango group	Active control	Mini-Balance Evaluation Systems Test (Mini-bestest) of Dynamic Balance Balance Evaluation—Systems Test	12 weeks	Rios Romenets et al.	[34]
The tango group improved in balance while the exercise group did not	Exercise group	Berg Balance Scale	20 tango Classes and 1 week before and 1 week after	Hackney et al.	[126]
Significant positive changes in balance in both groups	Partnered and non-partnered tango	Tandem Stance and One Leg Stance tests	20 tango classes	Hackney et al.	[45]
Gait					
Improvement in comfortable forward and dual task walking velocities	Control group—no intervention	Gaitrite	12 months of tango practice	Duncan et al.	[73]
Improvement in backwards stride length	Control group—no intervention	Berg Balance Scale, six minute walk distance, and backward stride length	13 weeks	Hackney et al.	[4]
Improvement in comfortable and fast as possible walking velocities, cadence	Partnered to non-partnered tango	Berg Balance Scale	10 weeks	Hackney et al.	[45]
Freezing of gait					
Tango group reported less freezing after 12 months compared to baseline	Control group—no intervention	Freezing of Gait Questionnaire	1 year	Duncan et al.	[73]
Endurance					
Maintaining the same endurance for the tango group and worsening in the no intervention control groups	Control group—no intervention	Six Minute Walk Test	1 year	Duncan et al.	[73]
Improvements in endurance	Control group—no intervention	Six Minute Walk Test	13 weeks	Hackney et al.	[4]
Improvements in endurance	Self-directed exercise group	Timed Up and Go and Dual-task Timed Up and Go	12 weeks	Rios Romenets et al.	[34]
Upper extremity function					
Tango group had improvements in upper extremity and hand function	Control group—no intervention	Nine Hole Peg Test	1 year	Duncan et al.	[73]
Fatigue					
Patients in the tango group had modest borderline improvement of fatigue	Self-directed exercise group	Krupp Fatigue Severity Scale	12 weeks	Rios Romenets et al.	[34]
Quality of life					
Improvements in the scores of Parkinson’s Disease Questionnaire-39 Summary Index and in Mobility and Social Support compared to the other groups	Waltz/ foxtrot, Tai Chi and no intervention	Parkinson’s Disease Questionnaire-39 Summary Index and in Mobility and Social Support	20 adapted tango sessions	Hackney et al.	[31]
A non-statistically difference was found in the quality of life of tango group	Self-directed exercise group	Parkinson’s Disease Questionnaire-39	12 weeks	Rios Romenets et al.	[34]
The quality of life was improved in the patients from the tango group	Support group	MDS-Unified Parkinson’s Disease Rating Subscale	1 year	Foster et al.	[75]
Participation					
Total current participation enhanced in the tango group compared to the control group without intervention, with overall activity retention improving from 77% to 90% in the tango group.	Control group—no intervention	Activity Card Sort	1 year	Foster et al.	[75]
Clinical global impression of change					
Significant changes in favor of the tango group in comparison to the self-directed exercise group from the examiner’s perspective only	Self-directed exercise group	Clinical Global Impression of Change	12 weeks	Rios Romenets et al.	[34]

## Data Availability

Not applicable.

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
