# Peer review of "Benefits of Tango Therapy in Alleviating the Motor and Non-Motor Symptoms of Parkinson’s Disease Patients—A Narrative Review"

_brainsci, 2022, doi:10.3390/brainsci12040448_

Round 1

Reviewer 1 Report

The manuscript “Benefits of Tango therapy in alleviating the motor and non-motor symptoms of Parkinson’s disease patients – a narrative review” reviews the previous literature associated with tango and Parkinson’s disease between 1980-2022. In the final review, they have included 100 papers. Overall, the review is well written and reviews the most important literature.

Below are my detailed comments and suggestions.

  • It would be helpful for the reader if the authors would add a “Limitations” section in which they show the limitations of the reviewed studies. For example, it seems that many studies have had a small sample size. Also, instead of just summarizing the findings in the end, could the authors add a “Future directions” section to outline the essential steps that still need to be taken in this field/gaps in this research area.
  • Because the review covers more than 40 years, could the authors mention whether they saw an impact of the long period on the reviewed studies? For example, how have studies changed during this period, or whether the authors found anything that was initially thought to occur in the earlier literature but later proved wrong, etc.? Have there been any controversies in the field?
  • Page 3, line 122: What does this mean “is effective and effective [29]”? Please clarify.
  • Page 3, line 115: What does this mean “especially [26] ”? Could the authors insert a word to describe [26]?

Author Response

Respected Reviewer,

We appreciate your assistance, advice, and support! We hope that our modifications have improved our ability to meet the expectations.

Reviewer 1

The manuscript “Benefits of Tango therapy in alleviating the motor and non-motor symptoms of Parkinson’s disease patients – a narrative review” reviews the previous literature associated with tango and Parkinson’s disease between 1980-2022. In the final review, they have included 100 papers. Overall, the review is well written and reviews the most important literature.

Below are my detailed comments and suggestions.

  • It would be helpful for the reader if the authors would add a “Limitations” section in which they show the limitations of the reviewed studies. For example, it seems that many studies have had a small sample size. Also, instead of just summarizing the findings in the end, could the authors add a “Future directions” section to outline the essential steps that still need to be taken in this field/gaps in this research area.

The sections Limitations and Future directions were added. Thank you for these insightful suggestions!

Lines: 714-755 “ 8. Further directions

            Existing research shows that tango is a beneficial method for people with Parkin-son's disease and positively affects PD-related symptoms. Additionally, significant im-provements in various motor and non-motor complaints have been seen. The impacts of tango on interpersonal interactions and psycho-emotional well-being have received relatively little attention. Additionally, the use of tango as a motivator in the treatment of patients with PD should be explored further. Scientific investigations should include a larger sample size and concentrate on long-term impacts.

The current research provides preliminary evidence for the usefulness of commu-nity-based dancing in alleviating the motor and non-motor symptomatology among people with Parkinson's disease. Future research might examine the nature of the changes in involvement that are happening, as well as the overall relevance of the many di-mensions of engagement to general health and well-being in people with Parkinson's disease.

We believe that more clinical trials on this subject are obviously needed. Ran-domised control studies must be developed and done together with neurologists, sports doctors, and investigators. Specific trial designs should be used to evaluate the potential disease-modifying benefits of tango therapy. Larger controlled studies assessing dance in Parkinson's Disease and moreover other types of dance therapies are required to es-tablish if certain dance intervention characteristics including dance style, abilities trained, amount of exercise, class length, are best suited for targeting specific symptoms in people with Parkinson's disease. Additional research with active control groups would be ben-eficial to determine tango's unique contribution in comparison to other exercise programs.

  1. Limitations

            Our narrative review primarily focused on the conclusions reached in various studies on the subject. The limited number of participants in each of the included studies is a significant limitation. The difficulty of identifying and integrating complicated connections which may occur across a vast number of research. Additionally, this review included studies involving patients with mild-to-moderate severity of illness. Studies including individuals with varying degrees of disease severity could enable investiga-tors to provide more personalized content and focus attention on whether tango gains in motor sign severity, life quality, equilibrium, mobility, and gait change throughout the illness's progression. The potential of making judgments that are misrepresentative is because of the selection bias and subjective assigning of the research selected for the evaluation. Also, comparing research literature is complicated by variations in research design, such as the program variables utilized including class duration, regularity, and period of the therapy, the kind of activity practiced, and the results assessed.

There are not enough dance investigations in the PD research that make compar-isons of dance therapies explicitly. To make recommendations to persons with PD about which activities would effectively address their symptoms and limitations while also promoting compliance, future research with bigger sample sizes will be necessary to evaluate different exercise programs.”

  • Because the review covers more than 40 years, could the authors mention whether they saw an impact of the long period on the reviewed studies? For example, how have studies changed during this period, or whether the authors found anything that was initially thought to occur in the earlier literature but later proved wrong, etc.? Have there been any controversies in the field?

Thank you for your suggestion, we have added the following lines:

Lines 684-712: “As our review examined articles published in the literature over a four-decade period, we noticed that studies conducted from the 1980s to the 2000s were modest in coverage, given the materials and methods used, which included more inventories and case reports, and we also noticed that the studies mentioned used subjectivist (related more on the answers of the patients and less on the neurological assessment) pre- and posttest scales and questionnaires as measuring instruments [i.e. 38, 87, 90]. Additionally, in relation to the aforementioned studies, we observed a dearth of scientific foundations for comparison, particularly in the introduction and discussion chapters. Because this is a relatively new field of study, studies conducted prior to the 2000s lacked the necessary foundation for analyzing the findings of articles in comparison to other publications in the literature to ascertain similarities or differences in the results obtained. Another feature that is more or less accurate is that research conducted in the beginning of the studied period had a more critical stance toward the issue due to a lack of evidence in the literature from that period.

Controversies in this field of scientific study and on this subject arise from the belief that tango had more of a placebo effect than an apparent impact on motor symptoms, fact reinforced by the use of self-assessment scales throughout the first investigations. Addi-tionally, the therapeutic implements for Parkinson's disease patients with non-motor symptoms were discovered later and initially received less attention because the non-motor symptomatology of Parkinson's disease patients was not adequately consid-ered, and the effects of non-motor symptomatology were not discovered to have such a significant impact on the patients' quality of life Another point of controversy stems from the fact that recovery through exercise, physiotherapy, and kinesiotherapy is extremely difficult and the effects decrease over time without consistent and continuous practice in this chronic and progressive neurodegenerative disease, whereas tango practice has demonstrated rapid and beneficial effects (after more than 20 tango lessons) sustained for an extended period of time. The comprehensive investigations on the subject of tango benefits in Parkinson's disease demonstrated the effectiveness of this alternative therapy over a considerable length of time of extensive studies.

  • Page 3, line 122: What does this mean “is effective and effective [29]”? Please clarify.

We have modified the phrase. “Balance enhancement is regarded as one of the most critical outcomes for determining if a treatment is effective and efficient [29].” Thank you!

  • Page 3, line 115: What does this mean “especially [26] ”? Could the authors insert a word to describe [26]?

We have modified the phrase. “Falling, stumbling, and trouble rotating as a consequence of postural instability are found to be significant [26].” Thank you!

Reviewer 2 Report

In my opinion, the article is written correctly, but the procedure for selecting the papers for analysis could be described in more detail. It is true, that it includes a critical analysis of publications on a specific problem, but unlike a systematic review, it is not carried out according to a strictly defined research protocol. Works of this type (narrative review) are burdened with a high degree of subjectivity. 

It is important that the authors do not give an unambiguous opinion. It is possible to use tango in the process of improving people with PD  - is it enough? 

In conclusion, the authors put a lot of work into preparing this article, presented a number of publications but the results are expected. 

Author Response

Respected Reviewer,

We appreciate your assistance, advice, and support! We hope that our modifications have improved our ability to meet the expectations.

Reviewer 2

In my opinion, the article is written correctly, but the procedure for selecting the papers for analysis could be described in more detail. It is true, that it includes a critical analysis of publications on a specific problem, but unlike a systematic review, it is not carried out according to a strictly defined research protocol. Works of this type (narrative review) are burdened with a high degree of subjectivity. 

We have provided more information about the procedure of analysis of the papers and research protocol.

Lines 53-76: “Our analysis obtained articles from the following electronic scientific resources: PubMed, Google Scholar, Web of Science, and Science Direct. Between 1980 and 2022, pertinent English publications were discovered using the terms "tango", “tango dance therapy” and "Argentinian tango" in conjunction with "Parkinson's illness" and "Park-inson's disease symptoms." One hundred articles were chosen through database searches. The search was manually supplemented with references from current studies and case reports. The titles of the documents were checked for accuracy, and duplicates were omitted. In total, 215 articles were identified; 31 articles were excluded after full-text screening and 84 articles were excluded because they were duplicates. Publications from the scientific literature on tango and Parkinson's illness met the eligibility criteria.

Reference lists from identified literature were manually searched for completeness by the authors to confirm content relevant to tango dance therapy in Parkinson's disease patients. Studies were included in this review if they met the following eligibility criteria: (1) written in English; (2) type of article being an original article, review, or case report; (3) patients participating in the studies being diagnosed with Parkinson's disease; (4) no external intervention during the experiment of the study involved; (5) the lot of patients with Parkinson's disease only performing tango therapy as an alternative therapy at the time of the study; and (6) the patients with Parkinson’s disease included in the study to not be diagnosed with dementia and psychiatric disorders.

Studies were excluded if they met the following criteria: (1) citations and patents; (2) written in a language other than English; (3) abstract papers with no data; and (4) the alternative therapy to be any other type of dance therapy other than tango or Argentinian tango. The search results were analyzed, and the important material is given in this paper as a narrative review.”

It is important that the authors do not give an unambiguous opinion. It is possible to use tango in the process of improving people with PD  - is it enough? 

In conclusion, the authors put a lot of work into preparing this article, presented a number of publications but the results are expected. 

We have added the following section: lines: 655-712. Thank you for your insightful suggestions!

“7. Results

As a result of our review, it was observed that tango is beneficial in Parkinson’s disease for reducing motor symptom severity and freezing of gait, improving balance, gait, endurance, and upper limb mobility. Furthermore, the non-motor symptoms have been beneficially influenced by tango regarding the reduction of fatigue, enhancing the quality of life, participation, and clinical global impression of change. Tango, in addition to medical therapy, is a useful tool in improving the lives of Parkinson’s disease patients, regarding both motor and non-motor symptoms, not to mention the other advantages non-Parkinson's disease-related of using this therapy.

Effects

In comparison with

Scale used

Duration of the study

Author

Bibliography

Motor symptom severity

Improvement in motor symptom severity after practicing tango

Control group – no intervention

The motor component of the Movement Disorder Society United Parkinson’s Disease Rating Scale Part III (MDS-UPDRS-III)

One year

Duncan et al

[73]

Improvements in motor symptom severity, improve balance, and functional mobility after practicing tango

Physiotherapy group or exercise group

The motor component of the Movement Disorder Society United Parkinson’s Disease Rating Scale Part III (MDS-UPDRS-III)

12 weeks

Rios Romenets et al

[34]

Stagnation for the symptomatology of the tango group, and worsening in the no intervention control group

Control group- no intervention

The motor component of the Movement Disorder Society United Parkinson’s Disease Rating Scale Part III (MDS-UPDRS-III)

Hackney et al

[4]

Balance

Improvements in balance among tango group

Control group- no intervention

Mini-Balance Evaluation Systems Test (Mini-bestest) of Dynamic Balance

One year

Duncan et al

[73]

Improvements in balance among tango group

Control group- no intervention

Mini-Balance Evaluation Systems Test (Mini-bestest) of Dynamic Balance

Two years

Duncan et al

[74]

Improvements in balance among tango group

Active control

Mini-Balance Evaluation Systems Test (Mini-bestest) of Dynamic Balance

Balance Evaluation—Systems Test

12 weeks

Rios Romenets et al

[34]

The tango group improved in balance while the exercise group did not

Exercise group

Berg Balance Scale

20 tango

Classes and one week before and one week after

Hackney et al

[126]

Significant positive changes in balance in both groups

Partnered and non-partnered tango

Tandem Stance and One Leg Stance tests

20 tango classes

Hackney et al

[45]

Gait

Improvement in comfortable for- ward and dual task walking velocities

Control group- no intervention

Gaitrite

12 months of tango practice

Duncan et al

[73]

Improvement in backwards stride length

Control group- no intervention

Berg Balance Scale, six minute walk distance, and backward stride length

13 weeks

Hackney et al

[4]

Improvement in comfortable and fast as possible walking veloc- ities, cadence

Partnered to non-partnered tango

Berg Balance Scale

10 weeks

Hackney et al

[45]

Freezing of gait

Tango group reported less freezing after 12 months compared to baseline

Control group- no intervention

Freezing of Gait Questionnaire

One year

Duncan et al

[73]

Endurance

Maintaining the same endurance for the tango group and worsening in the no intervention control groups

Control group- no intervention

Six Minute Walk Test

One year

Duncan et al

[73]

Improvements in endurance

Control group- no intervention

Six Minute Walk Test

13 weeks

Hackney et al

[4]

Improvements in endurance

Self-directed exercise group

Timed Up and Go and Dual-task Timed Up and Go

12 weeks

Rios Romenets et al

[34]

Upper extremity function

Tango group had improvements in upper extremity and hand function

Control group- no intervention

Nine Hole Peg Test

One year

Duncan et al

[73]

Fatigue

Patients in the tango group had modest borderline improvement of fatigue

Self-directed exercise group

Krupp Fatigue Severity Scale

12 weeks

Rios Romenets et al

[34]

Quality of life

Improvements in the scores of Parkinson’s Disease Questionnaire-39 Summary Index and in Mobility and Social Support compared to the other groups

Waltz/ foxtrot, Tai Chi and no intervention

Parkinson’s Disease Questionnaire-39 Summary Index and in Mobility and Social Support

20 adapted tango sessions

Hackney et al

[135]

A non-statistically difference was found in the quality of life of tango group

Self-directed exercise group

Parkinson’s Disease Questionnaire-39

12 weeks

Rios Romenets et al

[34]

The quality of life was improved in the patients from the tango group

Support group

MDS-Unified Parkinson's Disease Rating Subscale

One year

Foster et al

[75]

Participation

Total current participation enhanced in the tango group compared to the control group without intervention, with overall activity retention improving from 77% to 90% in the tango group.

Control group- no intervention

Activity Card Sort

One year

Foster et al

[75]

Clinical global impression of change

Significant changes in favor of the tango group in comparison to the self-directed exercise group from the examiner’s perspective only

Self-directed exercise group

Clinical Global Impression of Change

12 weeks

Rios Romenets et al

[34]

Table 1. Representative studies for the beneficial evolution of both motor and non-motor features of Parkinson’s disease after tango therapy.

Our narrative review is valuable because it summarizes the literature and provides direction, given that the sources are of appropriate methodological quality. Our narrative review expands the body of knowledge on the subject of Parkinson's disease therapy through tango and it's benefits being summarized after the literature citations were chosen to be appropriate and balanced.

Tango therapy is very successful because of the systematic examination and application of fundamental tango components. Furthermore, tango treatment is neither a straightforward physical therapy and neither a straightforward dance. Until today, it was commonly assumed that the advantages of tango therapy were psychological in nature. On the contrary, medical records indicate that the purpose of tango treatment is physical rehabilitation. The neuromuscular dysfunction is the most serious medical condition associated with Parkinson's disease which is alleviated in practicing tango. Tango therapy, on the other hand, has been found to have psychological positive benefits and has been used in psychiatric domains. Tango therapy provides both physical and psychological benefits for patients suffering from any condition, and it is more than a dance.

Tango therapy is indeed a confluence of different physical therapeutic interventions, as well as music and cognitive therapy. Tango therapy's results, nevertheless, appear to be greater than the total of numerous physical treatment modalities.

As our review examined articles published in the literature over a four-decade period, we noticed that studies conducted from the 1980s to the 2000s were modest in coverage, given the materials and methods used, which included more inventories and case reports, and we also noticed that the studies mentioned used subjectivist (related more on the answers of the patients and less on the neurological assessment) pre- and posttest scales and questionnaires as measuring instruments [i.e. 38, 87, 90]. Additionally, in relation to the aforementioned studies, we observed a dearth of scientific foundations for comparison, particularly in the introduction and discussion chapters. Because this is a relatively new field of study, studies conducted prior to the 2000s lacked the necessary foundation for analyzing the findings of articles in comparison to other publications in the literature to ascertain similarities or differences in the results obtained. Another feature that is more or less accurate is that research conducted in the beginning of the studied period had a more critical stance toward the issue due to a lack of evidence in the literature from that period.

Controversies in this field of scientific study and on this subject arise from the belief that tango had more of a placebo effect than an apparent impact on motor symptoms, fact reinforced by the use of self-assessment scales throughout the first investigations. Additionally, the therapeutic implements for Parkinson's disease patients with non-motor symptoms were discovered later and initially received less attention because the non-motor symptomatology of Parkinson's disease patients was not adequately considered, and the effects of non-motor symptomatology were not discovered to have such a significant impact on the patients' quality of life Another point of controversy stems from the fact that recovery through exercise, physiotherapy, and kinesiotherapy is extremely difficult and the effects decrease over time without consistent and continuous practice in this chronic and progressive neurodegenerative disease, whereas tango practice has demonstrated rapid and beneficial effects (after more than 20 tango lessons) sustained for an extended period of time. The comprehensive investigations on the subject of tango benefits in Parkinson's disease demonstrated the effectiveness of this alternative therapy over a considerable length of time of extensive studies.